# Moiré metrology of energy landscapes in van der Waals heterostructures

Dorri Halbertal [1✉], Nathan R. Finney [2], Sai S. Sunku[1], Alexander Kerelsky[1], Carmen Rubio-Verdú[1], Sara Shabani[1], Lede Xian [3,9], Stephen Carr[4,10], Shaowen Chen [1,11], Charles Zhang[1,12], Lei Wang [1,13], Derick Gonzalez-Acevedo[1,11], Alexander S. McLeod [1], Daniel Rhodes[1,14], Kenji Watanabe [5], Takashi Taniguchi[5], Efthimios Kaxiras [6], Cory R. Dean [1], James C. Hone [2], Abhay N. Pasupathy [1], Dante M. Kennes [3,7], Angel Rubio [3,8] & D. N. Basov [1]

The emerging field of twistronics, which harnesses the twist angle between two-dimensional materials, represents a promising route for the design of quantum materials, as the twist-angle-induced superlattices offer means to control topology and strong correlations. At the small twist limit, and particularly under strain, as atomic relaxation prevails, the emergent moiré superlattice encodes elusive insights into the local interlayer interaction. Here we introduce moiré metrology as a combined experiment-theory framework to probe the stacking energy landscape of bilayer structures at the 0.1 meV/atom scale, outperforming the gold-standard of quantum chemistry. Through studying the shapes of moiré domains with numerous nano-imaging techniques, and correlating with multi-scale modelling, we assess and refine first-principle models for the interlayer interaction. We document the prowess of moiré metrology for three representative twisted systems: bilayer graphene, double bilayer graphene and H-stacked $MoSe_2/WSe_2$. Moiré metrology establishes sought after experimental benchmarks for interlayer interaction, thus enabling accurate modelling of twisted multilayers.

[1] Department of Physics, Columbia University, New York, NY, USA. [2] Department of Mechanical Engineering, Columbia University, New York, NY, USA. [3] Max Planck Institute for the Structure and Dynamics of Matter and Center Free-Electron Laser Science, Luruper. Chaussee 149, 22761 Hamburg, Germany. [4] Department of Physics, Harvard University, Cambridge, MA 02138, USA. [5] National Institute for Material Science, Tsukuba, Japan. [6] John A. Paulson School of Engineering and Applied Sciences, Harvard University, Cambridge, MA 02138, USA. [7] Institut fur Theorie der Statistischen Physik, RWTH Aachen University, 52056 Aachen, Germany. [8] Center for Computational Quantum Physics, Flatiron Institute, New York, NY 10010, USA. [9] Present address: Songshan Lake Materials Laboratory, Dongguan, Guangdong 523808, China. [10] Present address: Brown University, Providence, RI 02912, USA. [11] Present address: Department of Physics, Harvard University, Cambridge, MA 02138, USA. [12] Present address: Department of Physics, University of California at Santa Barbara, Santa Barbara, CA 93106, USA. [13] Present address: National Laboratory of Solid-State Microstructures, School of Physics and Collaborative Innovation Center of Advanced Microstructures, Nanjing University, Nanjing, China. [14] Present address: Department of Materials Science and Engineering, University of Winsconsin-Madison, Madison, WI 53706, USA. ✉email: dh2917@columbia.edu

Twisted van der Waals structures, such as twisted bilayer graphene[1-16] (TBG), twisted double bilayer graphene[17-19] (TDBG), and twisted transition-metal-dichalcogenides[20-27] are in the vanguard of quantum materials research[28-30]. The twist between the layers leads to large-scale periodic perturbations of stacking configurations, called a moiré superlattice. Because atomic layers in van der Waals (vdW) materials are not rigid but instead behave as deformable membranes, moiré suprelattices acquire additional attributes. As two atomic layers with a small relative twist angle come in contact, the atomic positions relax to minimize the total energy. Through the relaxation process domains of lowest-energy configurations form and become separated by domain walls of transitionary configurations[31-33] (Fig. 1a). The generalized stacking fault energy function (GSFE), which provides the energetic variations across different stacking configurations, is the fundamental property that describes relaxed vdW interfaces[31,34]. The GSFE is commonly calculated using density functional theory (DFT)[31,34]. Experimental techniques[35] to probe the GSFE are currently restricted to the stable lowest-energy configuration, and are very limited in energy resolution compared to the variability among theoretical descriptions.

Here we show that the generalized stacking fault energy function (GSFE) is encoded in fine details of the relaxed moiré super-lattice patterns at the low twist-angle limit. In particular, the shape of domains and domain walls networks, as well as domain wall width, abide by transitionary configurations beyond the lowest-energy stackings of the domains. More specifically, we distinguish between single and double domain walls (SDW and DDW). SDWs separate two distinct stacking configurations of a

moiré superlattice (for instance, ABCA [MM'] and ABAB [MX'] in the TDBG [for twisted H-stacked MoSe$_2$/WSe$_2$, or T-H-MoSe$_2$/WSe$_2$ for short] example of Fig. 1a). DDWs, formed from the collapse of two SDWs, separate identical phases (ABAB for TDBG and MX' for T-H-MoSe$_2$/WSe$_2$ in Fig. 1a). The formation and nature of DDWs result from attraction of SDWs as they are brought together (for instance, due to external or relaxation induced strain), and is proven here to provide a reliable read-out of the underlying energetics. In cases of inequivalent two lowest-energy configurations (as in Fig. 1), the SDW develops a finite curvature $\kappa$, allowing one to extract the domains energy imbalance with an accuracy outperforming the ~3 meV/atom of the gold standard of quantum chemistry[36,37].

## Results

Moiré metrology, presented here, correlates measurable spatial patterns of the relaxed moiré superlattice (such as shapes of domains, SDWs and formation of DDWs) with modeling based on the GSFE. To do so, we developed a continuous two-dimensional relaxation simulation. The model searches for local interlayer displacement fields that minimize the total energy of the multilayer, as a sum of elastic and stacking energy terms (see Supplementary Information S1–2 for more details, also see ref. [33] for an alternative approach). The equations are solved in real space and thus capture subtle experimental details that remained underexplored. Fig. 1b–g is a tour-de-force of moiré metrology combining experimental imaging of different systems, techniques and length-scales (Fig. 1b–d), and their respective modeling

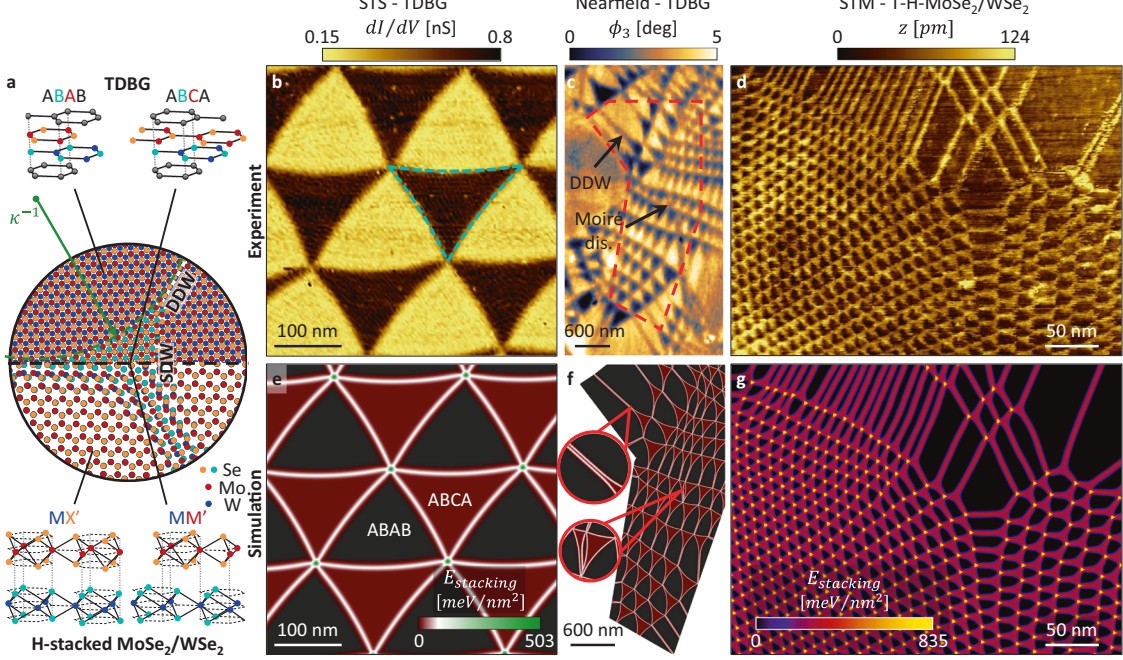

**Fig. 1 Physics of atomic layers stacking probed by moiré metrology in vdW twisted bilayers—experiment and theory. a** Illustration of domain formations in a relaxed twisted bilayer structure. Center: atomic positioning after relaxation (see Supplementary Information S1 for more details). Atoms are colored to highlight stacking configurations. The energy imbalance leads to curved single domain walls (SDWs), with radius of curvature indicated by $\kappa^{-1}$, and in some cases with formation of double domain walls (DDWs). Two systems with energy imbalance are considered: TDBG (top) and T-H-MoSe$_2$/WSe$_2$ (bottom). **b** STS map of TDBG with $\theta = 0.07°$, revealing rhombohedral (ABCA—dark) and Bernal (ABAB—bright) domains with minimal external strain. The rhombohedral phase bends inward (dashed turquoise line) revealing an energy imbalance between the two phases as discussed in the text. **c** Mid-IR (940 cm$^{-1}$) nearfield phase imaging of TDBG resolves ABCA (dark) and ABAB (bright) phases and DDW formations. **d** STM map of T-H-MoSe$_2$/WSe$_2$ resolving MM' (bright) and MX' (dark) stacking configurations as well as DDW formation in various strain conditions. **e–g** Stacking energy density from full relaxation calculations of the experimental cases of (**b-d**), respectively (see "Methods", Supplementary Information S1–2 and text for more details). The color-map is shared for (**e, f**). Magnified regions in **f** (and arrows in **c**) highlight a DDW formation and a moiré dislocation (see Discussion in Supplementary Information S3). Calculated region of (**f**) is marked by dashed shape in (**c**).

(Fig. 1e–g). Fig. 1b–d were acquired with modern scanning probe microscopy (SPM) techniques: scanning tunneling microscopy (constant current mode) and spectroscopy[17] (STM and STS respectively) and mid-infrared range (mid-IR) scanning nearfield optical microscopy[11] (SNOM). These techniques resolve stacking configurations based on local topographic and electronic (STM and STS), as well as mid-IR optical conductivity (mid-IR SNOM) contrasts. In low strain TDBG, the model (Fig. 1e) captures the fine curving of SDWs (Fig. 1b). In cases of higher strain (Fig. 1c and modeling in Fig. 1f) we observe the formation of one dimensional DDW structures (inset of Fig. 1f highlights an example). Similarly, DDW formations and SDW curving were observed (Fig. 1d) and modeled (Fig. 1g) in T-H-MoSe₂/WSe₂, with excellent agreement across different length-scales of the image (see Supplementary Information S3 for additional analysis). Next, we will illustrate in detail how moiré super-lattices reveal the energy landscape information using TBG and TDBG as prototypical examples.

**Moiré metrology of twisted bilayer graphene.** To study the energy landscape of TBG, we focus on the interplay between SDW and DDW formations. Fig. 2a presents a non-local nano-photocurrent map of TBG in the minimal twist limit <0.1°. Bright spots in the photo-current map highlight the AA sites (indicating higher absorption—see "Methods" and Supplementary Information S4). The AA sites are connected by domain walls separating AB and BA domains. The resultant moiré super-lattice is clearly affected by strain, inferred from the distorted triangular pattern, especially near the edges of the stack. There, we observe the merging of two SDWs into a single DDW (selected locations are marked in Fig. 2a). We successfully account for the observed

network within a model addressing a competition between SDWs and DDWs. To grasp the essential physics, we first assume a characteristic energy of forming a segment of DDW and SDW. We define a dimensionless domain-wall formation ratio as the ratio of DDW and SDW line energies, $\bar{\beta} = \gamma_{DDW}/\gamma_{SDW}$. In addition to $\bar{\beta}$, the model input includes the AA sites of the moiré pattern as the fixed vertices of the triangles forming the network. We explore the SDW vs. DDW structures that emerge for a given value of the single tuning parameter $\bar{\beta}$. The case of $\bar{\beta} = 2$ implies there is no benefit in forming a DDW, and the optimal structure would simply be straight SDWs connecting the AA sites. For $\bar{\beta} < 2$ the two SDWs attract each other favoring the emergence of DDW segments (see Supplementary Information S4 and Supplementary Video 1 for details). Our modeling captures the overall shape of the experimental map for $\bar{\beta} = 1.90$ (Fig. 2a). The agreement is remarkable considering the minimal modeling we employ. We conclude that in order for a TBG model to reproduce the experimental picture, two SDWs have to sufficiently attract one another as quantified by the fitted $\bar{\beta}$. In that sense, as we show more rigorously below, moiré metrology puts constraints the GSFE.

To quantify how the observed moiré networks constrain the stacking energy landscape, we span all realistic GSFE's satisfying the symmetry of TBG over a 2D unit-less parameter space $(\zeta, \tau)$ (as illustrated in Fig. 2b and discussed at Supplementary Information S4), such that each point on the $(\zeta, \tau)$ plane represents one GSFE candidate. We solve a set of 1D relaxation problems describing the profiles of SDWs and DDWs (see Supplementary Information S1 for more details on relaxation codes), and extract the domain wall formation ratio $\bar{\beta}$. This allows us to define a band in $(\zeta, \tau)$ plane of GSFE's that comply with the

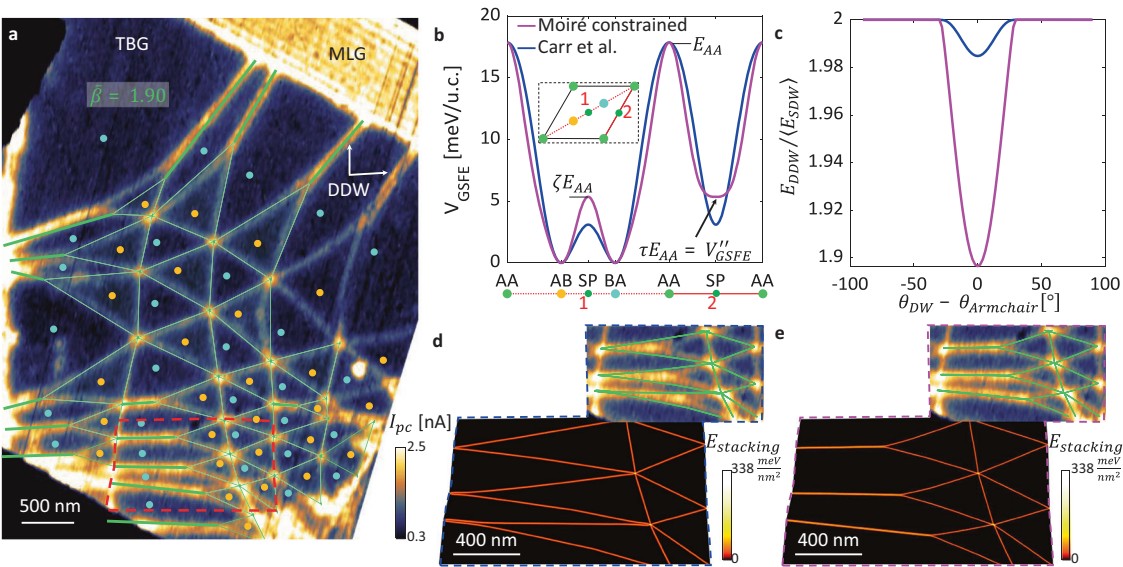

**Fig. 2 Energy landscape of twisted bilayer graphene (TBG) revealed by the interplay between double (DDW) and single (SDW) domain walls. a** Non-local nano-photocurrent map of moiré super-lattice of a TBG sample at the minimal twist limit (see "Methods" for more details). The technique reveals the formation of DDWs (marked by "DDW") at strained domains, separating domains of identical stacking configurations (each configuration is indicated by dots of a given color [AB—orange, BA—cyan]). The green network overlaid on the data corresponds to the prediction by a single tuning parameter model with $\bar{\beta} = 1.90$ (see text and Supplementary Information S6). **b** GSFE of TBG from ref. [31] (blue) and a moiré constrained version (magenta). The unit-less parameters $\zeta$, $\tau$, spanning the phase space of GSFE candidates for TBG, are illustrated (see Supplementary Information S6). Inset: path in configuration space for presented GSFE line-cuts. **c** Effective attraction between SDWs as reflected by DDW to SDW energy ratios for different SDW orientations (relative to armchair direction) for both models. $E_{DDW}$ is the DDW line-energy for a DDW along the armchair direction and similarly $E_{SDW}$ is for the average of the two SDWs comprising the DDW. **d, e** Stacking energy density from 2D relaxation calculation (see "Methods" and Supplementary Information S2) for the two discussed GSFE choices (**d**—literature, **e**—moiré constrained version with $\tau = 0.025$, $\zeta = 0.3$) of region marked by red dashed frame in (**a**), showing fundamental differences in formation of DDWs. Inset: extracted domain wall structures from relaxation calculations overlaid on experimental results.

experimental $\bar{\beta} = 1.90$ (see Supplementary Fig. S3g). Fig. 2b compares one GSFE moiré constrained candidate with the $\bar{\beta} = 1.90$ band (magenta) with the well-accepted choice of GSFE of ref. [31] (blue), which notably falls outside of the band with $\bar{\beta} = 1.98$. The moiré metrology analysis indicates that SDWs implied by GSFE in ref. [31] insufficiently attract one another (blue curve in Fig. 2c) to account for the observed network, as indeed revealed in Fig. 2d. In contrast, the moiré-constrained candidate (magenta in Fig. 2b) with a flatter saddle point promotes stronger SDWs attraction across a broad range of domain wall orientations (magenta curve in Fig. 2c; see Supplementary Information S1 and S5 for additional details), and yields excellent agreement with the data (Fig. 2e). Regardless of the good agreement, the relatively flat saddle point comes as a surprise, and may in fact correct for an unknown effect unrelated to interlayer energy.

**Moiré metrology of twisted double bilayer graphene.** Compared to TBG, the TDBG system makes an even more interesting case-study due to the small yet finite imbalance between the two lowest-energy phases: Bernal (ABAB) and rhombohedral (ABCA) stackings[17]. This imbalance results in an energy cost per-unit-area ($\sigma$) for rhombohedral relative to Bernal stackings, leading to characteristic curved domains[17,38] (see Fig. 1a, b). Exploring large areas of TDBG reveals a rich distribution of rhombohedral

domain shapes (see Fig. 1c and other TDBG images in this work). Fig. 3a summarizes this distribution as a histogram of inverse curvature values ($\kappa^{-1}$), extracted from images as in Fig. 1b (see Supplementary Information S6 for more examples). The histogram reveals a distinct clustering about a value of $\kappa^{-1} = 440 \pm 120$ nm, which we use to assess the accuracy of several variants of the GSFE from available DFT functionals (Fig. 3b and see "Methods"). All reported GSFE variants are qualitatively similar to the TBG case, peaking at the BAAC configuration, and having a saddle point barrier between ABAB and ABCA. A closer inspection (inset) reveals a profound difference between the GSFEs for the ABCA relative to the ABAB that governs domain curvature. We model the domain curvature and structure by a continuous 2D relaxation code (Supplementary Information S1). Fig. 3c, d show two representative cases, with disparate outcomes. In Fig. 3c (resembling the experimental case of Fig. 1b) the energy is minimized by slight bending of the SDW into the ABCA region. As the twist angle decreases (or as strain increases as in Fig. 1c), at some point it becomes energetically beneficial to form DDWs (Fig. 3d). As the twist angle further decreases, the shape of the ABCA domains remains unchanged. Similarly, solving for the domain formation for all DFT approaches and across a wide twist angle range, we compare the extracted $\kappa^{-1}$. Interestingly, $\kappa$ is independent of the twist angle for all GSFE variants (with values indicated by colored lines over Fig. 3a), which is not generally the

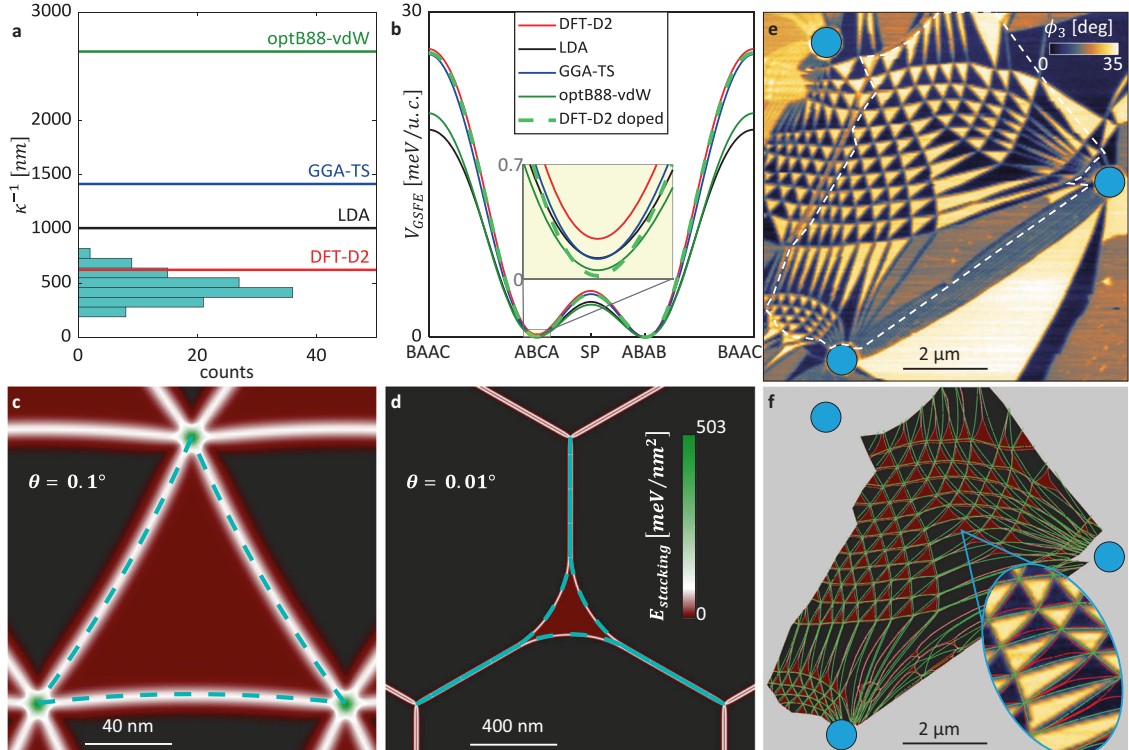

**Fig. 3 Moiré super-lattice study of rhombohedral domains in twisted double bilayer graphene (TDBG). a** Curvature histogram across all measured domains near charge neutrality point (see discussion in Supplementary Information S7), showing a clear cluster at 440 ± 120 nm. Calculated curvatures of 4 DFT approaches (of **b**) are illustrated over the histogram (colored horizontal lines). **b** GSFE of TDBG based on four different approaches (solid lines, see "Methods"). The dashed light-green line is the GSFE for DFT-D2 approach at a doping level of $8 \times 10^{12}$ cm$^{-2}$. Inset: enlarged view highlighting small difference between ABCA and ABAB configurations ($V_{GSFE}(ABAB) = 0$ identically). **c, d** Mechanical relaxation solutions (false-color: stacking energy density) and "soap-bubble" model domain shape (dashed turquoise) for two representative twist angles (**c**: $\theta = 0.1°$, **d**: $\theta = 0.01°$) for DFT-D2. **e** Mid-IR (940 cm$^{-1}$) nearfield phase imaging of a defect-induced doped TDBG (see Supplementary Information S7). 3 holes punctured one of the bilayers (blue circles—see "Methods") and induce a non-trivial external strain map. Mid-IR imaging resolves ABCA (dark) and ABAB (bright) phases and double-domain wall (DDW) formations (for instance, the multiple-DDW formation connecting bottom holes). **f** Comparing relaxation calculations solutions of un-doped vs. $8 \times 10^{12}$ cm$^2$ doped DFT-D2 approach GSFE, simulating the experimental case (marked by dashed shape in **e**), False-color represents stacking energy density of un-doped case, overlaid (green dots) with tracked domain walls in the doped case. Inset: highlighting differences between model by overlaying domain wall formation of doped (red) and un-doped (green) cases over strained region in the experimental map. The color-map is shared for (**c–e**).

case (see discussion in Supplementary Information S7). The domain structures are further captured by the 2D "soap-bubble" model, as seen in turquoise dashed lines in the representative cases of Fig. 3c, d and more generally in Supplementary Videos 2–5. This model approximates the total energy as a sum of a domain area term and two line-energy terms as $E = \int_{SDW} dl\gamma_1(\varphi) + \int_{DDW} dl\gamma_2(\varphi) + \sigma S$, where $\gamma_{1,2}$ are the line-energies of SDW and DDW as a function of the domain-wall orientation respectively, the integrations are along the domain walls, and $S$ is the area of the domain (see Supplementary Information S7). All model parameters require only the GSFE (and elastic properties) to describe domain shapes, with no additional tuning parameters (Supplementary Information S7). One approach, DFT-D2, remarkably reproduces the experimental cluster (Fig. 3a), due to relatively high $\sigma$ and comparable line-energies to other approaches (see Supplementary Information S2). It is noteworthy that the DFT-D approach has not been previously considered as the leading approach when theoretically benchmarked against the Quantum Monte-Carlo method for the binding energy of AA and AB stacking of bilayer graphene[39].

The rhombohedral domains represented in the histogram of Fig. 3a exemplify a well-defined electrostatic environment near charge neutrality point (CNP) in the absence of the interlayer bias. As shown recently[38], upon charging and biasing the balance between the rhombohedral and Bernal phases can shift. An extreme demonstration of malleability of TDBG moiré patterns under a non-uniform distribution of charges and high strain conditions is presented in Fig. 3e. Three holes (marked by blue circles) punctured one of the bilayers. This procedure prompts a highly strained moiré pattern, most strongly manifested in the densely packed parallel DDWs structures connecting the two bottom holes. The stack shows strong defect-induced doping (see discussion in Supplementary Information S5), apparent in the enhanced nearfield contrast between the ABCA (dark) and ABAB (bright) phases (compare to contrast of Fig. 1c). Further support for the high non-uniformity of charge distribution is an observed region of flipped balance, where the ABAB phase becomes unstable relative to ABCA across a sharp (~50 nm) interface (to the left of the top hole). Attempting to model the moiré superlattice with the DFT-D2 GSFE at CNP (red in Fig. 3b) fails to capture the observed structure of excessively curved SDWs (color-map of Fig. 3f). However, when introducing a doping level of $8 \times 10^{12}$ cm$^{-2}$ the resulting GSFE (dashed light-green in Fig. 3b) better captures the observed structure (green dots in Fig. 3f tracking the domain walls in the calculation). The difference between the two models becomes more pronounced for regions of higher strain, as highlighted in the inset of Fig. 3f (compare green and red dots in respect to the experimental map). Therefore, minuscule energy differences between models of order 0.1 meV/u.c. (inset of Fig. 3b) result in measurable spatial features of the relaxed moiré patterns. To put this figure in context, the theoretical method which is widely considered as the gold-standard of ab-initio quantum chemistry[36,37] yields an accuracy as low as 3 meV/atom[37].

## Discussion

To understand the enhanced sensitivity of moiré metrology under strain (as seen in Fig. 3f), we propose an alternative description of the moiré superlattice in terms of a geometric interference pattern of the lattices of the two layers (see Supplementary Information S1). At minute twist angles, the relaxed moiré patterns are essentially a projection of the detailed energy landscape over space, accumulated over large regions compared to the atomic scale. The introduction of strain between the layers, whether naturally occurring or externally controlled, alters the

interference pattern (Supplementary Information S8). As strain pushes the domain walls in Figs. 1–3 together, it also promotes their interaction; both effects are reflected in the relaxed moiré pattern (also see Supplementary Video 6).

Moiré metrology, introduced here, correlates first principle calculations of the stacking energy function with measurable spatial features of twisted vdW systems. The stacking energy function is widely used for modeling twisted multilayers across a broad range of twist angles and strain conditions, and has direct implications for the electronic band-structure[40]. Moiré metrology is not restricted to the discussed material systems and can be universally applied with other systems of great current interest[41–48]. Therefore, by providing a reliable account of the stacking energy function, moiré metrology has a broad impact across the field of vdW heterostructures. Furthermore, the moiré metrology tools can also be used for modeling and designing non-uniform strain fields in realistic devices. Finally, due to its outstanding stacking energy sensitivity, we propose moiré metrology as a concrete experimental path to provide much needed benchmarks for first-principle theoretical approaches[49].

## Methods

### Samples preparation

*Source crystals.* The bulk crystals of MoSe$_2$ and WSe$_2$ were grown by the self-flux method[50]: single crystals of MoSe$_2$ were prepared by combining Mo powder (99.997%, Alfa Aesar 39686) and Se shot (99.999%, Alfa Aesar 10603) in a ratio of 1:50 (Mo:Se) in a quartz ampoule. The ampoules were subsequently sealed under vacuum (~5 × 10$^{-6}$ Torr). The reagents were then heated to 1000 °C within 24 h and dwelled at this temperature for 8 weeks before being cooled to 350 °C over 4 weeks. At 350 °C the Se flux was decanted through alumina wool (Zircar D9202) in a centrifuge and the ampoules were quenched in air. The subsequently obtained MoSe$_2$ single crystals were annealed at 275 °C with a 200 °C gradient for 48 h to remove any residual Se. A similar process was also used to synthesize single crystals of WSe$_2$ with 1:15 (W:Se) as the starting ratio and using W powder (99.999%, Alfa Aesar 12973). Kish graphite source crystals were purchased from Graphene Supermarket.

*Exfoliation.* The MoSe$_2$ and WSe$_2$ monolayers, and Graphene and hBN flakes were mechanically exfoliated from the bulk single crystals onto SiO$_2$/Si (285 nm oxide thickness) chips using the tape-assisted exfoliation technique (the tape used was Scotch Magic Tape). The exfoliation followed ref. [51], such that the Si chips were treated with O$_2$ plasma (using a benchtop radio frequency oxygen plasma cleaner of Plasma Etch Inc., PE-50 XL, 100 W at a chamber pressure of ~215 mTorr) for 20 s for graphene, for 10 s for MoSe$_2$ and WSe$_2$, and no O$_2$ plasma treatment for hBN. The chips were then matched with respective exfoliation tape. In the graphene case the chip+tape assembly were heated at 100 °C for 60 s and cooled to room temperature prior to removing the tape. Such thermal treatment was not done for other materials. The MoSe$_2$ and WSe$_2$ monolayer relative crystallographic orientation was obtained by linear-polarization- resolved second-harmonic generation (SHG).

*Stack preparation.* All heterostructures were assembled using standard dry-transfer techniques[52] with a polypropylene carbonate (PPC) film mounted on a transparent-tape-covered polydimethylsiloxane (PDMS) stamp. The transparent tape layer was added to the stamp to mold the PDMS into a hemispherical shape which provides precise control of the PPC contact area during assembly[53].

All graphene heterostructures were made by first picking up the bottom-layer boron nitride (h-BN) (>25 nm thick), followed in the case of the TDBG samples by a graphite bottom gate-layer (>5 nm thick), then a dielectric BN layer (>25 nm thick). Prior to pick-up, mechanically exfoliated graphene flakes on Si/SiO$_2$ were separately patterned with anodic-oxidation lithography[54] to facilitate the "cut-and-stack" technique[55] where it was used (all samples but those used for STM, which used the established tearing method). In the case of the TDBG samples, additional anodic-oxidation lithography was used prior to pick-up to provide additional texture to the strain landscape, e.g., cut holes inside the bulk of one of the graphene layers or non-rectilinear edge geometries.

In the case of MoSe$_2$/WSe$_2$ the PPC was used to pick up a thin layer of exfoliated h-BN and a few layers of graphene. Then a monolayer WSe$_2$ was picked up and using the SHG data MoSe$_2$ monolayer was lifted on a rotation stage with ~1° twist angle. The stack was flipped over a Si/SiO$_2$ (285 nm) chip at 120 °C. In the last step, the sample was thermally annealed in a high vacuum chamber to remove the PPC at 250 °C for 1 h.

Some of the presented measurements (TDBG stacks of Fig. 3e, Supplementary Figs. S8b, d–f and red bins in histogram of Supplementary Fig. S8a) were taken at this point while the stack was on a PDMS/transparent-tape/PPC structure. This

provided access to the meta-stable large rhombohedral domain before they were suppressed by thermal annealing (see Supplementary Information S9).

After optional mid-assembly scanning probe measurement and/or optional encapsulation of the twisted-graphene layers, the PPC film with the heterostructure on top is mechanically removed from the transparent-tape-covered PDMS stamp and placed onto a Si/SiO$_2$ substrate such that the final pick-up layer is the top layer.

In the case of the TBG device presented in this work, the underlying PPC was removed by vacuum annealing at T = 350 °C. Standard plasma etching and metal deposition techniques[52] were then used to shape and make contact to the samples. In the case of the TDBG devices for STM (Fig. 1b) the stack was made using the established tearing method, using PPC as a polymer to sequentially pick up hBN, half of a piece of graphene followed by the second half with a twist angle.

For all samples for STM imaging, standard metal deposition techniques were avoided in order to maintain a pristine surface, therefore, direct contact was made to the stack by micro-soldering with Field's metal[56], keeping temperatures below 80 C during the entire process.

**Nearfield imaging techniques**. In this work, we used two nano-optical imaging techniques: cryogenic nano-photocurrent imaging (used for TBG imaging) and phase-resolved scattering type scanning optical microscope imaging (used for TDBG imaging) (s-SNOM). Cryogenic photocurrent imaging[57] was done with a home-built cryogenic SNOM, and s-SNOM nearfield imaging was done with a commercial (Neaspec) SNOM[11]. In both cases using mid-IR light (continuous wave CO$_2$ gas laser [Access Laser] at a wavelength of 10.6 μm) focused to a diffraction limited spot at the apex of a metallic tip, while raster scanning the sample at tapping mode. Fig. 2a was acquired at a temperature of 100 K and while tuning the silicon back-gate to a relatively high doping of $3 \times 10^{12}$ cm$^{-2}$. In such a case, we observe, for the first time, a non-local photo-current generation regime. In this regime, the light induced temperature profile is broad (relative to system size) and the photo-current generation is located at a distant interface (a monolayer twisted bilayer interface in this case, clearly visible as the bright region with plasmonic fringes at the top right section of Fig. 2a). The signal contrast in such a case results from absorption contrast between different stacking configuration, and not from thermo-electric properties. This unique approach provides a high-resolution image, not limited by thermal length-scales (see Supplementary Information S4 for more details).

In the s-SNOM case, we collect the scattered light (power of 3 mW) by a cryogenic HgCdTe detector (Kolmar Technologies). The far-field contribution to the signal can be eliminated from the signal by locking to a high harmonic (here we used the 3rd harmonic of the tapping). The phase of the backscattered signal was extracted using an interferometric detection method, the pseudo-heterodyne scheme, by interfering the scattered light with a modulated reference arm at the detector. Fig. 1c was acquired with a level of $1 \times 10^{12}$ cm$^2$ p-doping applied with a Si backgate. Such a level of doping has negligible effect on domain curvature (see Supplementary Information S5 for more details).

**STM and STS imaging**. STM and STS measurements were carried out in a home-built STM under ultra-high vacuum conditions. MoSe$_2$/WSe$_2$ measurements were performed at 300 K while TDBG measurements were performed at 5.7 K. The setpoints of the STM (constant current mode) imaging were V = −1.8 V and I = 100 pA for MoSe$_2$/WSe$_2$ measurements and V = 0.5 V and I = 50 pA for TDBG measurements. The tip-sample bias for the STS measurement of Fig. 1b was −57.5 meV. STM tips were prepared and calibrated for atomic sharpness and electronic integrity on freshly prepared Au (111) crystals. Samples were measured with multiple tips to ensure consistency of results.

**Parameter DFT calculations**. The DFT calculations of the GSFE parameters were performed with the Vienna Ab initio Simulation Package (VASP)[58]. Plane wave basis sets were employed with energy cutoff of 1200 eV and 500 eV for the calculations for TDBG and MoSe$_2$/WSe$_2$, respectively. Pseudopotentials were constructed with the projector augmented wave method (PAW)[59,60]. $60 \times 60 \times 1$ and $11 \times 11 \times 1$ Γ-centered k-point grids were used in the calculations for TDBG and MoSe$_2$/WSe$_2$, respectively. Vacuum spacing larger than 15 Å was added along the z direction to eliminate the artificial interaction between periodic slab images in all the calculations. In the calculations of the GSFE, the x–y coordinates of the atoms in all the 2D layers were fixed and the z coordinates were allowed to relax until forces in the z-direction are less than 1 meV/Å for TDBG and 20 meV/Å for MoSe$_2$/WSe$_2$. The van der Waals interactions are important in evaluating the energetics of the 2D layer structures. We tested this effect for TDBG by considering four approaches: (1) employing PAW pseudopotentials with the exchange-correlation functionals treated at the local density approximation (LDA) level. (2) Employing PAW pseudopotentials with the exchange-correlation functionals treated at the generalized gradient approximation (GGA) level[61] and additional van der Waals corrections are applied with the DFT-D2 method of Grimme[62]. (3) Employing PAW pseudopotentials with the GGA functionals and additional van der Waals corrections with the Tkatchenko-Scheffler method[63]. (4) Employing PAW pseudopotentials with a non-local correlation functional (optB88-vdW)[64–66] that approximately accounts for dispersion interactions. In the calculations for MoSe$_2$/WSe$_2$, we employed PAW pseudopotentials with the GGA PBE functionals

with the DFT-D3 van der Waals corrections[67]. The bulk modulus and shear modulus for each material were calculated by applying isotropic or uniaxial strain to a monolayer lattice and then performing a quadratic fit to the strain-dependent energies. For TDBG, we assume the elastic coefficients are twice the values extracted for monolayer graphene. The DFT ground state energy is evaluated on a regular grid of different interlayer configurations. The Fourier components of the resulting energies are then extracted to create a convenient functional form for the GSFE.

## Data availability

The raw datasets used for the presented analysis within the current study are available from the corresponding author on reasonable request.

## Code availability

Developed relaxation codes can be provided from the corresponding author on reasonable request.

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

## Acknowledgements

Research at Columbia on moiré superlattices is supported as part of Programmable Quantum Materials, an Energy Frontier Research Center funded by the U.S. Department of Energy (DOE), Office of Science, Basic Energy Sciences (BES), under award DE-SC0019443. STM instrumentation for STM experiments was developed with support from the Air Force Office of Scientific Research via grant FA9550-16-1-0601. Synthesis of MoSe2 and WSe2 crystals was supported by the NSF MRSEC program through Columbia in the Center for Precision Assembly of Superstratic and Superatomic Solids (DMR-2011738). A.R. acknowledges support by the European Research Council (ERC-2015-AdG-694097), Grupos Consolidados (IT1249-19), SFB925 and the Flatiron Institute, a division of the Simons Foundation. We acknowledge funding by the Deutsche Forschungsgemeinschaft (DFG) under Germany's Excellence Strategy - Cluster of Excellence Matter and Light for Quantum Computing (ML4Q) EXC 2004/1—390534769, funding by Advanced Imaging of Matter (AIM) EXC 2056—390715994, funding by the DFG under RTG 1995 and RTG 2247 and by DFG within the Priority Program SPP 2244 "2DMP". We acknowledge support from the Max Planck-New York City Center for Non-Equilibrium Quantum Phenomena. Work at Harvard was supported by the STC Center for Integrated Quantum Materials NSF Grant No. DMR1231319 and ARO MURI Award No. W911NF-14-0247. D.H. was supported by a grant from the Simons Foundation (579913). N.R.F. acknowledges support from the Stewardship Science Graduate Fellowship program provided under cooperative agreement number DE-NA0003864. C.R.V. acknowledges funding from the European Union's Horizon 2020 research and innovation programme under the Marie Skłodowska-Curie grant agreement No 844271. D.N.B. is the Vannevar Bush Faculty Fellow ONR-VB: N00014-19-1-263 and the Moore Investigator in Quantum Materials EPIQS #9455.

## Author contributions

D.H. conceived the moiré metrology framework, developed the relaxation codes and performed related analysis. S. Chen and D.G.A. prepared the TBG device with supervision by J.C.H. and C.R.D. N.R.F. and C.Z. fabricated all TDBG devices for nano-optics experiments with supervision by J.C.H. and C.R.D. L.W. fabricated TDBG devices for STM experiments with supervision by J.C.H. S.S. fabricated T-H-MoSe2/WSe2 devices and performed their STM study with supervision by A.N.P. D.R. grew the MoSe2 and WSe2 crystals with supervision by J.C.H. K.W. and T.T. grew the hBN crystals. Cryogenic nano-photocurrent imaging of TBG was done by D.H. and S.S.S. with assistance from A.S.M. and supervision by D.N.B. D.H. performed s-SNOM imaging of TDBG with supervision by D.N.B. A.K. and C.R.V. performed and analyzed the STM/STS measurements of TDBG with supervision by A.N.P. L.X. performed DFT calculations for GSFE of TDBG with supervision by A.R. and S.Carr performed DFT calculations for GSFE of MoSe2/WSe2 with supervision by E.K. Manuscript was written by D.H. and D.N.B. with contributions by D.M.K., N.R.F., S.Carr, A.S.M., and A.R. All authors contributed by discussions with particular contribution with interpretation of theory by D.M.K, S.Carr and A.R.

## Competing interests

The authors declare no competing interests.
