## [Peer Review File · Nature Communications]

Reviewers' Comments:

Reviewer #1:

Remarks to the Author:

This manuscript reports very interesting observations on the structure of local stacking in moiré superlattices in twisted two-dimensional materials. This work is one of the first where the energetics of different stacking configurations, distinguished by the orientation of domain walls between Bernal (for graphene) and 2H (for transition metal dichalcogenides) stacking domains has been reported.

I recommend this manuscript for the publication, upon several minor technical corrections.

1. In Fig. 1a, the choice of colours for the background (yellow used for both in the sketch of the lattice is confusing, as the top view of the lattice in the nominally 2H stacking domains does not correspond to 2H stacking.

2. In the theory part, several DFT+vdW codes have been used. It would be good to know which of those are in the best agreement with the Quantum Monte Carlo (QMC) modelling of interlayer adhesion in Bernal bilayers [PRL 115, 115501 (2015)], which was recently used for benchmarking of DFT-vdW functionals. It would also be good to provide references to the sources/papers where those functionals have been reported. From the first glance, the one that the authors give the highest credit is from the DFT-D family, where the earlier used functional produced widely spread results, far from the QMC-set mark.

Reviewer #2:

Remarks to the Author:

The manuscript "Moiré metrology of energy landscapes in van der Waals heterostructures" by Dorri Halbertal and co-workers present a very appealing approach, combining different nano-imaging techniques with ab-initio calculations, to gain a deeper insight into the heterogeneous energy landscape present in moiré superlattices in 2D materials stacks. These systems are clearly a hot topic in nanoscience after the discovery of unconventional superconductivity in twisted bilayer graphene. Since that observation there are still too many open questions and this work can shed some light to answer some of them. I thus anticipate the huge impact of this work in the community working on 2D materials and thus I recommend the publication of this work asap. The text is clear, the data carefully presented. I do have some minor comments that might improve the manuscript:

1) page 3, line 55. the authors claim that the STM measurements provide topographic information. While this is the case for samples with homogeneous electronic properties, this might be far from reality in a sample where the density of states is varying spatially or in samples where the local barrier height is inhomogeneous. In the case of a moiré superlattice it is clear that the STM cannot provide pure topographic information.

2) The materials and method section is too vague. There are no references to the source materials used for the synthesis. No description of the synthesis at all (temperature, time, ...). No description of the mechanical exfoliation (e.g. which tape you used?). Please provide a much more accurate description of the materials and methods in order to help the readers to reproduce your results in the future, this should be a crucial point in science.

Reviewer #3:

Remarks to the Author:

In the submitted manuscript, the authors present experimental and theoretical methodology allowing for studies of energy landscapes in van der Waals heterostructures with twisted

constituent layers. The state-of-the-art experimental techniques together with the computational tools allow for extracting extremely intriguing physics. The so-called 'twistronics' is very hot topic nowadays and interesting phenomena are observed in the whole plethora of twisted, vertically stacked two-dimensional materials.

In my opinion, the submitted manuscript is fairly novel and of considerable importance in the emerging field. The paper is written clearly and in comprehensive way. It is supplemented by details of the work, and very good movies demonstrating the dependence of the phenomena on parameters. Therefore, I would like to recommend the publication of the submitted manuscript. However, I have two points that might be addressed by the authors.

i) The authors cited solely the literature dealing with graphene and transition metal dichalcogenides. However, in 2019 and 2020 there have appeared many papers on twisted 2D layers dealing with materials beyond the two mentioned above. For example, the paper on twisted bilayers of GeSe in Nature Communications (<https://doi.org/10.1038/s41467-020-14947-0>) and some other. Maybe, just to stress the universality of methodology proposed by the authors, it wouldn't be bad to extend the reference list.

ii) In lines 320 - 327 of the manuscript, the authors mention 4 approaches for approximations for exchange and correlation functionals. In the listed approaches (1) and (2) they mentions pseudopotentials. My question is: Do the pseudopotentials (or PAWs) used in these 4 calculations were generated consistently employing identical exchange-correlation functional that in the calculations of the twisted layers, or identical pseudo potential was used in all 4 calculations.? This requires clarification.

Point-by-point response to the reviewers' comments

Response in bold font below.

Reviewer #1 (Remarks to the Author):

This manuscript reports very interesting observations on the structure of local stacking in moiré superlattices in twisted two-dimensional materials. This work is one of the first where the energetics of different stacking configurations, distinguished by the orientation of domain walls between Bernal (for graphene) and 2H (for transition metal dichalcogenides) stacking domains has been reported.

I recommend this manuscript for the publication, upon several minor technical corrections.

We thank the reviewer for his kind words and for the expressed support of our work.

1. In Fig. 1a, the choice of colours for the background (yellow used for both in the sketch of the lattice is confusing, as the top view of the lattice in the nominally 2H stacking domains does not correspond to 2H stacking.

The original intention was to schematically convey in a single panel the different sub-components of the moiré superlattice for the systems under discussion. In particular: The two lowest energy configurations, single and double domain walls and the curvature of the domain wall due to phase energy imbalance. Due to the inherent differences between the TDBG and 2H-MoSe₂/WSe₂ systems, in doing so the atomic description of one of the systems was not appropriately captured by the color scheme and atomic positions in the central schematic of Fig. 1a. We understand the reviewer's point, and agree that it may be misleading. We therefore revised Fig. 1a to better represent the configurations of both systems. We do so by splitting the central panel into a top (TDBG) and bottom (2H-MoSe₂/WSe₂) sections. The bottom section now includes a dedicated simulation for the 2H-MoSe₂/WSe₂ system (previously they both shared the same simulation). Consequently, we also revised the caption of Fig. 1 and the relaxation simulation description in Supplementary Information section S2.

2. In the theory part, several DFT+vdW codes have been used. It would be good to know which of those are in the best agreement with the Quantum Monte Carlo (QMC) modelling of interlayer adhesion in Bernal bilayers [PRL 115, 115501 (2015)], which was recently used for benchmarking of DFT-vdW functionals. It would also be good to provide references to the sources/papers where those functionals have been reported. From the first glance, the one that the authors give the highest credit is from the DFT-D family, where the earlier used functional produced widely spread results, far from the QMC-set mark.

We benchmarked four approaches of DFT calculations with the experimental results for twisted double bilayer graphene (TDBG), including three types of exchange-correlation functionals, namely, LDA, GGA (plus D2 or TS vdW corrections), and opt88-vdW functionals. Within these approaches, we found that the DFT(GGA)+D2 method matches the best with the experimental results (see Fig. 3). We therefore highlighted this approach as the one offering best agreement with the experimental observations, regardless of insight by strictly theoretical benchmarking. We note that according to the mentioned theoretical benchmark [PRL 115, 115501 (2015)], LDA seems to have the best agreement with the QMC modelling in calculating the binding energy (BE) for AB stacking bilayer graphene. One should note,

though, that the benchmarking was done with only two particular stacking configurations (AA and AB) for bilayer graphene with a fixed lattice constant in the QMC calculations.

It is quite common that some vdW methods do well in certain situations but fail in others. For example, the same reference shows that vdW+DF method (and not LDA) is in the best agreement with the QMC method in calculating the BE for the AA stacking bilayer graphene.

Our results are not necessarily in contrast with the theoretical benchmarking in PRL 115, 115501 (2015) because of the different target quantities (BE in certain stacking vs. overall energy landscape at different stacking) and systems (bilayer vs. double bilayer). Ideally, the best vdW method to be developed should pass both the experimental and the theoretical benchmarking.

In our work we make the claim of the need to introduce moiré superlattice based experimental benchmarking in addition to the more conventionally used theoretical benchmarking.

We have included all the supporting references (references 58-67 in the revised manuscript) for the DFT-vdW approaches in the Methods section.

Reviewer #2 (Remarks to the Author):

The manuscript "Moiré metrology of energy landscapes in van der Waals heterostructures" by Dorri Halbertal and co-workers present a very appealing approach, combining different nano-imaging techniques with ab-initio calculations, to gain a deeper insight into the heterogeneous energy landscape present in moiré superlattices in 2D materials stacks. These systems are clearly a hot topic in nanoscience after the discovery of unconventional superconductivity in twisted bilayer graphene. Since that observation there are still too many open questions and this works can seed some light to answer some of them. I thus anticipate the huge impact of this work in the community working on 2D materials and thus I recommend the publication of this work asap. The text is clear, the data carefully presented.

We are pleased the reviewers shares the excitement about the potential impact of this work and is in favor of publication.

I do have some minor comments that might improve the manuscript:

1) page 3, line 55. the authors claim that the STM measurements provide topographic information. While this is the case for samples with homogeneous electronic properties, this might be far from reality in a sample where the density of states is varying spatially or in samples where the local barrier height is inhomogeneous. In the case of a moiré superlattice it is clear that the STM cannot provide pure topographic information.

The reviewer is of course correct to point this out. The term 'STM topography' is commonly used in STM literature synonymously with the constant tunneling current mode. However, as the reviewer pointed out this mode contains a convolved topography and DOS information. We added a clarification on the matter in the first paragraph of the Results section and removed the use of the word 'topography' where not needed to avoid any confusion.

2) The materials and method section is too vague. There are no references to the source materials used for the synthesis. No description of the synthesis at all (temperature, time, ...). No description of the mechanical exfoliation (e.g. which tape you used?). Please provide a much more accurate description of the materials and methods in order to help the readers to reproduce your results in the future, this should be a crucial point in science.

We thank the reviewer for this important point. We made revisions to the Methods section to provide better clarity about fine experimental details. We believe this provides a fuller description that allows others to reproduce our results.

Reviewer #3 (Remarks to the Author):

In the submitted manuscript, the authors present experimental and theoretical methodology allowing for studies of energy landscapes in van der Waals heterostructures with twisted constituent layers. The state-of-the-art experimental techniques together with the computational tools allow for extracting extremely intriguing physics. The so-called 'twistronics' is very hot topic nowadays and interesting phenomena are observed in the whole plethora of twisted, vertically stacked two-dimensional materials.

In my opinion, the submitted manuscript is fairly novel and of considerable importance in the emerging field. The paper is written clearly and in comprehensive way. It is supplemented by details of the work, and very good movies demonstrating the dependence of the phenomena on parameters. Therefore, I would like to recommend the publication of the submitted manuscript.

We thank the reviewer for the support of our work and its publication in Nature Communications.

However, I have two points that might be addressed by the authors.

i) The authors cited solely the literature dealing with graphene and transition metal dichalcogenides. However, in 2019 and 2020 there have appeared many papers on twisted 2D layers dealing with materials beyond the two mentioned above. For example, the paper on twisted bilayers of GeSe in Nature Communications (<https://doi.org/10.1038/s41467-020-14947-0>) and some other. Maybe, just to stress the universality of methodology proposed by the authors, it wouldn't be bad to extend the reference list.

We fully agree with the reviewer that the presented methodology is universal, and is not restricted to the discussed material systems. This is indeed one of the main take-home messages of the manuscript. We extended the reference list (added references are highlighted) to include more systems of interest, ranging from other TMDCs homo and hetero-bilayer structures, GeSe, hBN, 2D magnets and correlated oxides. A statement about the universality of the concept was also added at the discussion section following this request.

ii) In lines 320 - 327 of the manuscript, the authors mention 4 approaches for approximations for exchange and correlation functionals. In the listed approaches (1) and (2) they mention pseudopotentials. My question is: Do the pseudopotentials (or PAWs) used in these 4 calculations were generated consistently employing identical exchange-correlation functional that in the calculations of the twisted layers, or identical pseudo potential was used in all 4 calculations.? This requires clarification.

We thank the referee for pointing it out. For all 4 calculations of TDBG we have used the same PAW pseudopotentials, but three different types of exchange-correlation functionals, namely LDA, GGA, and opt88-vdW. With GGA functionals, we employed two types of vdW corrections, namely, DFT-D2 and DFT-TS. We have now revised the description in the Methods section to clarify this point

Reviewers' Comments:

Reviewer #1:

Remarks to the Author:

I have read the revised paper and the authors' response, and I am fully satisfied by their revision and arguments. I recommend to publish this manuscript.

Reviewer #2 (Remarks to the Author):

The authors have address the points raised in my referee report very accurately and I don't have further comments. I recommend the publication of the revised manuscript as it is.

Reviewer #3:

Remarks to the Author:

The authors addressed satisfactorily my queries, and in my opinion, also of other reviewers. Therefore, I would like to recommend the publication of the manuscript in its revised form.

Point-by-point response to the reviewers' comments

Response in bold font below.

Reviewer #1 (Remarks to the Author):

I have read the revised paper and the authors' response, and I am fully satisfied by their revision and arguments. I recommend to publish this manuscript.

We thank the reviewer for the kind words and for the expressed support of publication of our work in Nature Communications in its latest form.

Reviewer #2 (Remarks to the Author):

The authors have address the points raised in my referee report very accurately and I don't have further comments. I recommend the publication of the revised manuscript as it is.

We thank the reviewer for the kind words and for the expressed support of publication of our work in Nature Communications in its latest form.

Reviewer #3 (Remarks to the Author):

The authors addressed satisfactorily my queries, and in my opinion, also of other reviewers. Therefore, I would like to recommend the publication of the manuscript in its revised form.

We thank the reviewer for the kind words and for the expressed support of publication of our work in Nature Communications in its latest form.